# Dendrimers with Tetraphenylmethane Moiety as a Central Core: Synthesis, a Pore Study and the Adsorption of Volatile Organic Compounds

**DOI:** 10.3390/ijms231911155

**Published:** 2022-09-22

**Authors:** Zi-Ting Gu, Chung-Hao Tzeng, Hung-Jui Chien, Chun-Chi Chen, Long-Li Lai

**Affiliations:** 1Department of Applied Chemistry, National Chi Nan University, No. 1 University Rd., Puli, Nantou 54561, Taiwan; 2Department of Environmental Engineering and Science, Feng Chia University, Taichung 40724, Taiwan

**Keywords:** dendrimer, adsorption, VOC, flexible framework, void space

## Abstract

Reasonable yields of two dendrimers with central tetraphenylmethane and peripheral *3*,*5*-di-(*tert*-butanoylamino)benzoylpiperazine moieties are prepared. These dendrimers have a void space in the solid state so they adsorb guest molecules. Their BET values vary, depending on the H-bond interaction between the peripheral moiety and the gas molecules, and the dendritic framework that fabricates the void space is flexible. In the presence of polar gas molecules such as CO_2_, the BET increases significantly and is about 4–8 times the BET under N_2_. One dendrimer adsorbs cyanobenzene to a level of 436 mg/g, which, to the authors’ best knowledge, is almost equivalent to the highest reported value in the literature.

## 1. Introduction

Dendrimers with a central core, bridging and peripheral units have been the subject of many studies and are produced using a convergent or divergent approach that allows their monodispersity and control of the molecular size [1]. Different functional groups are incorporated at the periphery or internal part, and this produces special desired properties. A long flexible chain that is attached at the periphery promotes the formation of mesogenic phases of dendrimers [2,3,4,5,6,7]. A rigid or semi-rigid moiety may act as a linker for the fabrication of intramolecular void spaces, so many dendrimers adsorb metal ions or small molecules in solution. In addition to liquid crystals [2,3,4,5,6,7], these marcromolecules have therefore been focused on drug delivery [8,9,10] or catalysis [11,12,13], too.

Like metal-organic frameworks (MOFs) [14,15,16], dendrimers have a void space in the solid state and adsorb gas molecules or volatile organic compounds (VOCs) [17,18,19,20]. However, dendrimers do not generally contain metal ions in their framework and are always constructed with C-C, C-N or C-O covalent bonds, so they have a low density and good thermal stability. Most dendrimers are also soluble in organic solvents and can be reprocessed and purified easily, so they are more suited to practical applications than MOFs. In particular, dendrimers with non-flexible frameworks are noteworthy as they can contain void space in the solid stacking, which can be used for sensing guest molecules as shown by Müllen [21,22]. The non-flexible framework may also prevent dendrimers from close packing, which may result in quenching, so the rigid dendrimers, studied in light harvesting by Balzani, are very interesting [23,24]. 

A previous study by the authors designed *3*,*5*-di-(*tert*-butanoylamino)benzoylpiperazine (BABP-H) as a peripheral block for constructing dendrimers and showed that H-bond interaction allows the dendrimers to fabricate a void space in the solid state [19]. In the continuing research, we aim at the relative applications and further prepare dendrimers with BABP units, as shown in Figure 1, for adsorbing VOCs, such as benzonitrile, which is a major undesired component of biofuel and generates nitrogen oxides after combustion, thus causing acid rain [25]. 

To promote the adsorption of VOCs, a tetraphenyl moiety is used as a central core, and dendrimers **TAPM-4Den** and **TAPM-8Den** are produced. Both dendrimers are soluble in organic solvents, so the adsorbed VOCs are easily quantified using ^1^H-NMR spectroscopy, which is more convenient than using gravimetric gas analysis [15] or the traditional GC technique [26,27]. According to the literature, the greatest adsorption capacity for benzonitrile is 442 mg/g [16]. The present study shows that the adsorption capacity for **TAPM-8Den** is 436 mg/g, which is almost equivalent to the best-reported value in the literature.

## 2. Results and Discussion

*3*,*5*-di-(*tert*-butanoylamino)benzoylpiperazine (**BABP-H**) was produced using the method of a previous study [19]. Eight equivalents of **BABP-H** were treated with **TAPM-8Cl** in the presence of triethylamine in THF at 80 °C in a sealed tube to produce the dendrimer, **TAPM-8Den** with a yield of 60%. Four equivalents of **BABP-H** were reacted with **TAPM-8Cl** in the presence of triethylamine in THF at 0 °C, generating the dendrimer, **TAPM-4 Den,** with a yield of 65% (Figure 1). **TAPM-8 Den** and **TAPM-4Den** were characterized using elemental analysis, mass spectrometry and ^1^H- and ^13^C-NMR spectroscopy.

The mass spectrum in Figure 1 was obtained using MALDI-TOF and shows that there is a peak of m/z at 3788.1 from the [M]^+^ ion of **TAPM-8Den**. A peak of m/z at 2380.0 from [M]^+^ ion of **TAPM-4Den** was also observed (Appendix A). The dendrimer, **TAPM-8Den,** is stable up to 400 °C, as shown by TGA analysis (Figure 2). The slight weight loss at less than 200 °C is due to water vaporization because amide moieties at the peripheral part of the dendrimer adsorb water molecules from the surroundings, which is also supported by the results of elemental analysis. The thermal stability of **TAPM-4Den** is slightly less than that of **TAPM-8Den**, possibly due to the reactive chloro substituents in **TAPM-4Den** (Appendix A). The error between experimental and theoretical values for **TAPM-8Den**·12H_2_O and **TAPM-4Den**·6H_2_O for C, H and N is less than 0.4%.

The dendrimers, **TAPM-8Den** and **TAPM-4Den,** were first degassed under vacuum at 110 °C for 3 h to remove moisture, and the adsorptions of CO_2_ gas at 195K and N_2_ gas at 77 K were studied (Figure 3). The Brunauer–Emmett–Teller (BET) surface areas of **TAPM-8Den** and **TAPM-4Den** were calculated using CO_2_ isotherms at 195 K as 213.7 and 177.6 m^2^/g, respectively, using the method of a previous study [28,29,30]. 

The BET surface area of **TAPM-8Den** and **TAPM-4Den** is calculated to be 27.0 and 45.9 m^2^/g, respectively, using the N_2_ isotherms at 77 K. The void spaces in both dendrimers due to the sorption of CO_2_ are larger than those due to the sorption of N_2_ sorption because the CO_2_ isotherms were measured at 195 K and the N_2_ isotherms were measured at 77 K. At higher temperatures, the molecular stacking is less condensed, and the corresponding void space increases. However, the BET from CO_2_ isotherms is about eight times that of the BET from N_2_ isotherms for **TAPM-8Den,** and the BET from CO_2_ estimation is about four times that of the BET from N_2_ estimation for **TAPM-4Den**. Using the CO_2_ isotherms, the BET for **TAPM-8Den** is greater than that for **TAPM-4Den**, but using the N_2_ isotherms, the BET for **TAPM-8Den** is less than that of **TAPM-4Den**. 

Based on the above observation of isotherms, it is proposed that the frameworks of dendrimers **TAPM-8Den** and **TAPM-4Den** are flexible. If there is less interaction between the dendrimer and the gas (for example, the interaction between the amido moiety of dendrimer and N_2_ is not significant), the size of the fabricated void space depends on the stacking between dendrimers. One molecule of **TAPM-4Den** has four **BABP** moieties but one molecule of **TAPM-8Den** has eight **BABP** moieties. During stacking, the amido groups in the **TAPM-8Den** moiety strongly interact with each other by H-bonding, so **TAPM-8Den** is more condensed and has a lower BET. 

N_2_ isotherms were used to calculate the pore sizes for **TAPM-8Den** and **TAPM-4Den**. Figure 4 shows that the pore distribution for **TAPM-8Den** is 13.5–71.0 Å and most are between 13.5 and 20.0 Å. The pore distribution for **TAPM-4Den** is 16.0–105.0 Å and most are between 25.0 and 70.0 Å. The pores of **TAPM-8Den** are generally smaller than those of **TAPM-4Den** because there is more H-bond interaction between **BABP** moieties of **TAPM-8Den**.

If there is stronger interaction between the dendrimer and the gas molecules (for example, the interaction between the amido moiety of dendrimer and the O of CO_2_ is significant), the void space that is fabricated by the dendritic framework increases in size because gas is introduced between dendritic molecules. Therefore, the BET of **TAPM-8Den** is greater than that of **TAPM-4Den**. The TEM images of **TAPM-8Den** and **TAPM-4Den,** taken under similar conditions, are shown in Figure 5; the shapes of both dendrimers look somewhat circular, and the pores of two dendrimers seem to be amorphous.

The sorptions of CO_2_ by **TAPM-8****Den** and **TAPM-4****Den** at 273 and 298 K (Appendix A) are used to calculate the isosteric heat for CO_2_ sorption (*Q*_st_) using the virial method (Appendix A) [31,32,33,34] The *Q*_st_ value for **TAPM-8****Den** was measured as 26.5 kJ/mol at zero coverage, and the *Q*_st_ value for **TAPM-4****Den** is 27.2 kJ/mol at zero coverage (Figure 6). Both *Q*_st_ values gradually decrease when more CO_2_ is adsorbed by dendrimers **TAPM-8****Den** and **TAPM-4****Den**.

For an uptake of 0.1 mmole, the *Q*_st_ value for **TAPM-8****Den** is similar to that for **TAPM -4****Den**, which is ~25.0 kJ/mol. For an uptake of less than 0.1 mmole, the *Q*_st_ value for **TAPM-4****Den** is a little greater than that for **TAPM-8****Den**. For an uptake over 0.1 mmole, the *Q*_st_ value for **TAPM-4****Den** is less than that for **TAPM-8****Den**. The *Q*_st_ value for **TAPM-4****Den** decreases much faster than that for **TAPM-8****Den** when more CO_2_ is adsorbed by porous dendrimers. 

For a very low uptake of CO_2_, the pores of **TAPM-8Den** and **TAPM-4Den** are similar in size to those under N_2_ because the void space that is fabricated by the dendritic framework arises from the H-bond interaction between **BABP** moieties [19]. Therefore, CO_2_ is adsorbed by **TAPM-4Den** more quickly than by **TAPM-8Den**, so the *Q*_st_ value for **TAPM-4Den** is greater at this stage. If more CO_2_ is adsorbed, the interaction of CO_2_ with the **BABP** moiety of dendrimers is more important, so the void space that is fabricated by the dendritic framework increases in size because CO_2_ is inserted between **BABP** moieties. 

**TAPM-8****Den** contains more **BABP** moieties at the periphery so the corresponding interaction in **TAPM-8****Den** is much more significant. Therefore, **TAPM-8****Den** has more void spaces and the *Q*_st_ value decreases more slowly (Figure 6).

As the H-bond interaction plays an important role in fabricating the void space of **TAPM-8****Den** and **TAPM-4****Den**, both compounds were investigated by FT-IR spectroscopy. As indicated in the literature, the free N-H stretching and hydrogen bond N-H stretching of the amide moiety arises at about 3444 cm^−1^ and 3310 cm^−1^, respectively [35,36]. **TAPM-8****Den** and **TAPM-4****Den** consist of a strong stretching at about 3303 cm^−1^ and 3332 cm^−1^, respectively (Appendix A), indicating the existence of the strong H-bond interaction of **TAPM-8****Den** and **TAPM-4****Den** in their solid stackings.

To support the assumption that the framework of **TAPM-8****Den** and **TAPM-4****Den** is flexible, both dendrimers were used to adsorb VOCs. The dendrimer, **TAPM-8****Den** or **TAPM-4****Den**, was firstly recrystallized from hexane-THF (3:1), and ^1^H-NMR spectroscopy was used to confirm the complete removal of solvents under vacuum at 110 °C for 3 h. 

The vacuumed **TAPM-8****Den** or **TAPM-4****Den** (6.5 mg) was kept in a closed bottle (50 mL) containing a small amount of VOC (1 mL) in an isolated bottle (2 mL) at room temperature (~28 °C). After 24 h, **TAPM-8****Den** or **TAPM-4****Den** was removed from the VOC bottle and placed in a fume hood for 10 min. Samples containing VOC were then dissolved in DMSO-D_6_ for a ^1^H-NMR spectroscopic analysis, and a specific chemical shift (δ) of the dendrimer, away from the signals for H_2_O and other impurities, was used as an internal standard to calculate the amount of adsorbed VOC. 

The intensity at ~8.1 ppm due to the chemical shift of H (at C_4_) in the BABP moiety (Figure 1) was compared with the intensity for *o*-H (~7.8 ppm (d)) in cyanobenzene, and the amount of cyanobenzene that is adsorbed was thus measured. The chemical shifts for the dendrimers, **TAPM-8****Den** or **TAPM-4****Den** and cycnobenzene are all away from the signals for H_2_O and other impurities, so signal interference can be excluded. The amounts of other VOCs that are adsorbed by the dendrimers, **TAPM-8****Den** or **TAPM-4****Den,** were also identified in a similar manner. 

Table 1 shows that one molecule of the dendrimer, **TAPM-4****Den,** adsorbs 0.54, 4.07, 3.61, 1.37, 0.38, 0.42 and 0.38 equivalents of hexane, benzonitrile, nitrobenzene, toluene, *o*-xylene, *m*-xylene and *p*-xylene, respectively (Appendix A). One molecule of the dendrimer, **TAPM-8****Den,** adsorbs 0.34, 16.07, 9.73, 1.80, 1.92, 1.57 and 1.62 equivalents of the corresponding VOCs. 

Studies show that the capacity of porous materials to adsorb VOCs depends on the functionalities on the pore surface of the substrate and the void spaces that are fabricated by the dendritic framework and on the physical properties of VOCs, such as dipole moments, boiling point and kinetic diameter (Table 2) [37,38]. The kinetic diameter of the adsorbed VOCs is 4–8 Å, which is smaller than the pore sizes for the two dendrimers that are studied (Figure 4), so they can be adsorbed by **TAPM-4****Den** and **TAPM-8****Den**. The kinetic diameter of cyanobenzene and nitrobenzene are not reported in the literature but are believed to fall in this range, as their molecular size is comparable to that of toluene and xylene. 

Only 0.54 and 0.34 equivalents of hexane are, respectively, adsorbed by **TAPM-4****Den** and **TAPM-8****Den**. Hexane has a very small dipole moment (0.09 debye), so there is little interaction between hexane and the porous substrates of **TAPM-4****Den** and **TAPM-8****Den**. The void spaces that are fabricated by both dendrimers are almost unaffected by adsorption, so only a small amount of hexane is adsorbed by each dendrimer, and **TAPM-8****Den** adsorbs less hexane than **TAPM-4****Den**. This observation agrees with the results of N_2_ isotherms (Figure 3). 

When toluene, *o*-xylene, *m*-xylene and *p*-xylene are used as VOCs, the adsorption capacity of **TAPM-4****Den** remains virtually unchanged, even though the dipole moments of *o*-xylene and *p*-xylene are, respectively, slightly greater and less than that of hexane. However, the adsorption capacity of **TAPM-8****Den** increases (Table 2). *p*-Xylene has a zero dipole moment, but the **TAPM-8****Den** adsorbs more *p*-xylene than **TAPM-4****Den**, which may arise from the H-π interaction between the amido moiety of BABP and the benzene derivative. However, the corresponding interaction in **TAPM-4****Den** is much less significant. This is also true if cyanobenzene and nitrobenzene are used as VOCs because these two molecules exhibit strong polarity and interact strongly with the amido moiety of BABP in an H-bond interaction. 

**TAPM-8Den** has more BABP moieties, so the H-bond interaction has a more significant effect in **TAPM-8Den** than in **TAPM-4Den,** and significantly more cyanobenzene or nitrobenzene is adsorbed by **TAPM-8Den** than by **TAPM-4Den**, which is in agreement with the results of CO_2_ isotherm at 273 K (Figure 3). **TAPM-8Den** adsorbs more cyanobenzene than nitrobenzene, possibly because cyanobenzene has a lower boiling point than nitrobenzene, so cyanobenzene vaporizes more easily into the pores of dendrimers under similar conditions. 

The ^1^H-NMR spectroscopy results show that TAPM-8Den adsorbs ~16 equivalents of cyanobenzene (Appendix A), which is equivalent to 436 mg/g of the adsorption capacity. In adsorbing nitrogen-containing compounds, such as cyanobenzene, by porous materials, the H-bond interaction between the porous substrates and the N of VOCs should be very important, and several reports on removing nitrogen-containing compounds from biofuels by adsorptions have been published [16,43,44,45,46]. The best adsorption capacity for benzonitrile is 442 mg/g according to the literature [16], which is almost equivalent to our reported value although the BET value based on N_2_ isotherms for TAPM-8Den is smaller (27.0 m^2^/g) than the BET values for other porous MOFs [14,15,16]. This may be due to the function of a flexible dendritic framework.

## 3. Methods and Materials

### 3.1. General

With no further purification, all reagents were used as received. ^1^H and ^13^C NMR spectra were measured using a Bruker AMX-300 spectrometer. The mass spectra were recorded using a Microflex MALDI-TOF MS. Elemental analysis used an Elementar Unicube analyzer (EA000600). Thermogravimetric analysis used a Perkin-Elmer TGA-7 TG analyzer under N_2_. Brunauer–Emmett–Teller (BET) analysis used a Micrometrics TriStar II Plus system, with N_2_ as the adsorbate at 77 K and CO_2_ as the adsorbate at 195 K, 273 K and 298 K, respectively. All gases for these experiments had a purity of 99.9995%. The morphology of the samples was studied using high-resolution electron microscopy (HRTEM, JEOL JEM-2010) at 200 kV. 

### 3.2. Preparation of **TAPM-8Cl**

Cyanuric chloride (1.987 g, 10.775 mmol) was added to acetone (32 mL) in an ice bath, and tetrakis(4-aminophenyl)methane (1 g, 2.628 mmol) in acetone (21 mL) was then slowly added. The resulting solution was stirred at 0 °C for 1 h. After the reaction, K_2_CO_3_ (1.141 g, 10.770 mmol) in water (250 mL) was added to produce an intermediate, **TAPM-8Cl**, in 74.3% yield. As **TAPM-8Cl** decomposes gradually, it was immediately used for further reaction after preparation. ^1^H-NMR (300 MHz, DMSO-d_6_, 25 °C, TMS): δ = 7.19 (d, 8H, *J* = 8.7Hz, 8×Ar-H), 7.54 (d_,_ 8H, *J* = 8.7Hz, 8×Ar-H), 11.19 (s, 4H, 4×NH) ppm.

### 3.3. Preparation of **TAPM-8Den**

*3*,*5*-di-(*tert*-butanoylamino)benzoylpiperazine (BABP-H) (0.776 g, 2 mmol), prepared by the literature method [19], was dissolved in THF (40 mL) and then added to **TAPM****-8C****l** (0.243 g, 0.25 mmol) in THF (20 mL) at 0 °C. Triethylamine (1.2 mL, 6 mmol) was added, and the resulting mixture was heated at 80 °C in a sealed tube for 16 h. Solvent was removed at reduced pressure, and water (100 mL) was added to generate a solid, which was further purified by chromatography (SiO_2_: 2.1 × 15 cm; eluate: THF:CH_2_Cl_2_ = 1:1) to give a crude product. Pure **TAPM-8Den** was thus obtained in 60% yield (0.57g) after further recrystallization by THF-hexane (1;10). ^1^H-NMR (300 MHz, DMSO-d_6_, 25 °C, TMS): δ = 1.19(s, 144H, 48×CH_3_), 3.42~3.77(m_,_ 64H, 32×CH_2_), 7.00(s, 8H, 8×Ar-H), 7.39(s, 16H, 16×Ar-H), 7.61(s, 8H, 8×Ar-H), 8.09(s, 8H, 8×Ar-H), 9.15(s, 4H, 4×NH), 9.32(s, 16H, 16×NH) ppm; ^13^C-NMR (75 MHz, DMSO-d_6_, 25 °C, TMS): δ = 27.12, 41.63, 42.81, 47.19, 113.52, 113.95, 117.95, 130.64, 136.69, 137.91, 139.42, 163.99, 164.65, 169.04, 176.61 ppm; MS: M/Z: calcd for C_205_H_268_N_48_O_24_ (M)^+^: 3788.6; found: 3788.1; elemental analysis: calcd (%) for (C_205_H_268_N_48_O_24_ · 12H_2_O) C 61.48, H 7.35, N 16.79; found: C 61.57, H 7.42, N 16.79.

### 3.4. Preparation of **TAPM-4Den**

*3*,*5*-di-(*tert*-butanoylamino)benzoylpiperazine (BABP-H) (0.311g, 0.8 mmol) was dissolved in THF (20 mL) and then added to **TAPM-8Cl** (0.194 g, 0.2 mmol) in THF (20 mL) at 0 °C. Triethylamine (0.5 mL, 2.4 mmol) was added, and the resulting mixture was stirred at 0 °C for 2 h. Solvent was removed at reduced pressure, and water (100 mL) was added to generate a solid, which was further purified by chromatography (SiO_2_: 2.1 × 15 cm; eluate: THF-hexane (1;1) to give a crude product. Pure **TAPM-4Den** was thus obtained in 65% yield (0.31g) after further recrystallization by THF-hexane (1;10). ^1^H-NMR (300 MHz, DMSO-d_6_, 25 °C, TMS): δ = 1.19(s, 72H, 24×CH_3_), 3.42~3.77(m_,_ 32H, 16×CH_2_), 7.07(s, 4H, 4×Ar-H), 7.39(s, 8H, 8×Ar-H), 7.60(s, 8H, 8×Ar-H), 8.09(s, 8H, 8×Ar-H), 9.32(s, 8H, 8×NH), 10.18(s, 4H, 4×NH) ppm; ^13^C-NMR (75 MHz, DMSO-d_6_, 25 °C, TMS): δ = 27.06, 40.07, 43.15, 46.84, 47.50, 113.91, 118.45, 119.79, 125.06, 130.77, 131.22, 135.51, 136.43, 139.40, 141.24, 163.13, 163.64, 164.27, 168.11, 169.02, 176.52 ppm; MS: M/Z: calcd for C_121_H_144_Cl_4_N_32_O_12_ (M)^+^: 2380.5; found: 2380.0; elemental analysis: calcd (%) for (C_121_H_144_Cl_4_N_32_O_12_
**·** 6H_2_O) C 58.40, H 6.32, N 18.01; found: C 58.53, H 6.42, N 17.61.

## 4. Conclusions

The dendrimers, **TAPM-8Den** and **TAPM-4Den**, which have tetraphenylmethane as a central core and *3*,*5*-di-(*tert*-butanoylamino)benzoylpiperazine as a peripheral functionality, are successfully produced and are observed to contain void space in the solid state. These void spaces are fabricated by an H-bond interaction between BABP at the periphery, so the dendritic framework is regarded to be flexible when guest molecules are present inside the dendrimers. 

In the presence of more polar molecules, there is a stronger interaction between the dendrimer and guest molecules, so the H-bond interaction between BABP moieties may be weakened, and guest molecules insert between BABPs to expand the void space that is fabricated by the dendritic framework. In the presence of fewer polar molecules, there is a weaker interaction between dendrimer and guest molecules, so the H-bond interaction between BABP moieties is less changed, and the void space that is fabricated by the BABP moiety does not change significantly. The size of void spaces inside the dendritic framework depends on the degree of interaction between BABP and guest molecules. Therefore, although the BET value for the dendrimer, **TAPM-8Den,** is much smaller than that for other porous MOFs, **TAPM-8Den** has a good adsorption capacity (436 mg/g) because it has a flexible framework. Therefore, **TAPM-8Den** is a good candidate for adsorbing nitrogen-containing VOCs in the biofuel industry or the related laboratory. Particularly, **TAPM-8Den** can be prepared in a mild condition and easily recrystallized for removing VOCs after usage.

## Data Availability

The data presented in this study are available on request from the corresponding author.

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
