# Peer review of "Dendrimers with Tetraphenylmethane Moiety as a Central Core: Synthesis, a Pore Study and the Adsorption of Volatile Organic Compounds"

_ijms, 2022, doi:10.3390/ijms231911155_

Round 1

Reviewer 1 Report

The manuscript ijms-1898744 "Dendrimers with Tetraphenylmethane Moiety as a Central Core: Synthesis, a Pore Study and the Adsorption of Volatile Organic Compounds" by Lai and co-workers describes the synthesis of two dendrimers with central tetraphenylmethane and peripheral 3,5-di-(tert-butanoylamino)benzoylpiperazine fragments and the study of their capacity to adsorb some volatile organic compounds (hexane, benzonitrile, nitrobenzene, toluene, o-xylene, m-xylene and p-xylene). The authors have interesting experimental results, so I think that this paper will be of interest to the readers of IJMS.

Questions and comments:

1) Why did the authors not use an excess of the BABP-H reagent in the synthesis of the TAPM-8Den compound to increase the yield of the target compound?

2) How did the authors confirm the structure of the TAPM-4Den compound, namely, the substitution of one chlorine atom for each branching fragment (Cyanuric chloride fragment)?

3) Images of 1H, 13C NMR, IR spectra of novel compounds should be added in supplementary materials.

4) According to the pore size data determined by N2 isotherms, the obtained materials are very inhomogeneous. How can this be explained? Have the authors tried to improve their homogeneity?

5) Please note the signals of adsorbents (TAPM-4Den or TAPM-8Den) and VOCs in Figures S4.

6) The authors write that "H-bond interaction between BABP moieties" are observed. However, the presence of these hydrogen bonds is not confirmed by any direct physical method.

7) Have the authors tried to determine the adsorption capacity by other methods (not 1Н NMR)? How correct is it to use only one method to establish such relationships?

8) I recommend the authors to compare the obtained results with those known in the literature.

9) The authors need to add the information about the possible further application of the obtained results to the conclusions.

10) Minor changes:

- Please add VOC abbreviation transcript in Introduction.

- Lines 74, 89 and 104. Maybe 3,5-di-(tert-butanoylamino)benzoylpiperazine is BABP-H (not BABP)?

Author Response

Reviewer 1:

  • Why did the authors not use an excess of the BABP-H reagent in the synthesis of the TAPM-8Den compound to increase the yield of the target compound?

Author reply: Thanks for the comment. We did try to use excess of BABP-H reagent for the TAPM-8Den production. However, this make the purification process more difficult. By the present reaction ratio, TAPM-8Den was produced in an acceptable yield (60%), and can be purified using a simple recrystallization process.

  • How did the authors confirm the structure of the TAPM-4Den compound, namely, the substitution of one chlorine atom for each branching fragment (Cyanuric chloride fragment)?

Author reply: Thanks for the comment. The reaction temperature for substituting the first and second chloro moiety is quite different. As shown in the experiment, the first chloro substitution takes place at 0 °C very easily and the second chloro substitution only undergoes at 80 °C slowly. BABP-H has been allowed to react with 8 equivalents of TAPM-8Cl in THF in 0 °C over two hours, and then checked by mass spectrometry several times. The results showed that no trace of TAPM-5Den compound was detected.

  • Images of 1H, 13C NMR, IR spectra of novel compounds should be added in supplementary materials.

Author reply: Thanks for the comment. These spectra are added in supporting information as Figure S4 and Figure S5.

  • According to the pore size data determined by N2 isotherms, the obtained materials are very inhomogeneous. How can this be explained? Have the authors tried to improve their homogeneity?

Author reply: Thanks for the comment. The referee is correct. The obtained materials look very inhomogeneous, which may arise from the quick process of recrystallization (the product was dissolved in THF first and hexane was added to precipitate it). We did not improve their homogeneity but will try to do that by slowing down the recrystallization speed in the future work. 

  • Please note the signals of adsorbents (TAPM-4Den or TAPM-8Den) and VOCs in Figures S4.

Author reply: Thanks for the comment. It looks some signals of adsorbents after adsorbing VOCs are a little different from the signals of original adsorbents or VOCs. There are some interactions between adsorbents and VOC, which leads to the small variation of chemical shifts. In addition, the chemical shifts for calculating the adsorbed ratio were indicated in Figure S6 of the revised manuscript.

  • The authors write that "H-bond interaction between BABP moieties" are observed. However, the presence of these hydrogen bonds is not confirmed by any direct physical method.

Author reply: Thanks for the comment. The H-bond interaction is now confirmed by FT-IR study and mentioned in line 202-208 of the revised manuscript. The statement is as follows: As the H-bond interaction plays an important role in fabricate void space of TAPM-8Den and TAPM-4Den, both compounds were investigated by FT-IR spectroscopy. As indicated in the literature, the free N-H stretching and hydrogen bond N-H stretching of the amide moiety arises at about 3444 cm−1 and 3310 cm−1, respectively [31,32]. TAPM-8Den and TAPM-4Den consist of a strong stretching at about 3303 cm−1 and 3332 cm−1, respectively (Figure S4), indicating the existence of the strong H-bond interaction of TAPM-8Den and TAPM-4Den in their solid stackings

7)  Have the authors tried to determine the adsorption capacity by other methods (not 1Н NMR)? How correct is it to use only one method to establish such relationships?

Author reply: Thanks for the comment. We did not use other methods to determine the adsorption capacity of adsorbents, but it may be worth trying in the future work. By the 1H-NMR spectroscopy, the error percentage should be within 5%, as 5% of impurity is not easily detected by 1H-NMR spectroscopy.

  • I recommend the authors to compare the obtained results with those known in the literature.

Author reply: Thanks for the comment. Other works on removing nitrogen-containing compounds by adsorption was added in line L268-275 of the revised manuscript. The statement is as follows: In adsorbing nitrogen-containing compounds, such as cyanobenzene, by porous materials, the H-bond interaction between the porous substrates and the N of VOCs should be very important, and several reports on removing nitrogen-containing compounds from biofuels by adsorptions have been published [16, 35-38]. The best adsorption capacity for benzonitrile is 442 mg/g according to the literature [16], which is almost equivalent to our reported value although the BET value based on N2 isotherms for TAPM-8Den is smaller (27.0 m2/g) than the BET values for other porous MOFs [14-16].

  • The authors need to add the information about the possible further application of the obtained results to the conclusions.

Author reply: Thanks for the comment. The possible further application of TAPM-8Den is added in Conclusion (Line 294-297). The statement is as follow: Therefore, TAPM-8Den is a good candidate for adsorbing nitrogen-containing VOCs in the biofuel industry or the related laboratory. Particularly, TAPM-8Den can be prepared in a mild condition and easily recrystallized for removing VOCs after usage.

10) Minor changes:

- Please add VOC abbreviation transcript in Introduction.

- Lines 74, 89 and 104. Maybe 3,5-di-(tert-butanoylamino)benzoylpiperazine is BABP-H (not BABP)?

Author reply: Thanks for the comment. The VOC abbreviation transcript is added in Introduction (line 32) and BABP is changed to BABP-H in Lines 43, 79, 94, 109, 110 and 112 of the revised manuscript.

Reviewer 2 Report

Rigid dendrimers are an interesting topic worth investigating. However I am missing references to the large body of work from Klaus Müllen at the MPI on polyphenylenedendrimers, which were among the first dendrimers of this class. The polypyridyl systems of Balzani and coworkers is another system which could be referenced to here. It would also have been nice if a larger dendrimer has been synthesized in order to show the scope of the method.

Author Response

Referee 2

Rigid dendrimers are an interesting topic worth investigating. However I am missing references to the large body of work from Klaus Müllen at the MPI on polyphenylenedendrimers, which were among the first dendrimers of this class. The polypyridyl systems of Balzani and coworkers is another system which could be referenced to here. It would also have been nice if a larger dendrimer has been synthesized in order to show the scope of the method.

Author reply: Thanks for the comment. The works of Professor Müllen and Professor Balzani are added in Introduction (line 37-41). The statement is as follows: In particular, dendrimers with non-flexible framework are noteworthy, they can contain void space in the solid stacking, which can be used for sensing guest molecules as shown by Müllen [21,22]. The non-flexible framework may also prevent dendrimer from close packing that may result in quenching, so the rigid dendrimers, studied in light harvesting by Balzani, are very interesting [23,24].

Round 2

Reviewer 1 Report

I thank the authors for answering my questions and improving the manuscript.